# Exploring the Clinical Utility of the Music Therapy Assessment Tool for Awareness in Disorders of Consciousness (MATADOC) with People with End-Stage Dementia

**DOI:** 10.3390/brainsci12101306

**Published:** 2022-09-28

**Authors:** Wendy Louise Magee, Anne Wheeler Lipe, Takayoshi Ikeda, Richard John Siegert

**Affiliations:** 1Boyer College of Music and Dance, Temple University, Philadelphia, PA 19132, USA; 2Independent Researcher, Richfield, NC 28137, USA; 3Blue Earth Security Co., Ltd., Chuo-ku, Tokyo 104-0032, Japan; 4Department of Psychology and Neuroscience, School of Clinical Sciences, Auckland University of Technology, Auckland 0627, New Zealand

**Keywords:** music therapy, assessment, end-stage dementia, clinical utility, MATADOC, MiDAS

## Abstract

Dementia is a major health concern globally and cross-culturally with progressive decline in cognition, mobility and communication. There are few interventions for end-stage dementia (ESD) although music interventions have been observed to be accessible for people with mid to late-stage dementia. The lack of protocols and measures suited to ESD has limited research into the effects of music therapy. Measure sensitivity to minimal responsiveness is one limitation to the use of existing music intervention measures with ESD. This exploratory study examined the clinical utility of the Music Therapy Assessment Tool for Awareness in Disorders of Consciousness (MATADOC) for use with people with end-stage dementia, including preliminary reliability and validity. The MATADOC is a standardized assessment for minimally responsive patients with disorders of consciousness and may be useful for ESD. Using repeated measures with blinded MATADOC-trained raters, MATADOC data were collected with a small convenience sample of people with ESD in a residential care setting. Clinical utility data were collected from the raters and evaluated using a multidimensional model. To explore its functionality, MATADOC outcomes were compared to another measure for music interventions in dementia. The MATADOC may be useful for assessing functioning and responsiveness to music interventions for people with ESD without the risk of floor effects. Modifying the MATADOC protocol and assessment documentation prior to testing with a larger sample will enhance its sensitivity specific to ESD and age-related needs, providing a new music-based ESD assessment.

## 1. Introduction

### 1.1. Introduction to the Population: Dementia and End-Stage Dementia

The term *dementia* refers to a set of symptoms which include impaired cognitive functions affecting everyday living such as memory, language, problem-solving skills and executive functioning. While brain changes associated with dementia characterize a number of diseases, Alzheimer’s is the most common form of dementia accounting for between 60 and 80% of cases [1]. It is estimated that 6.2 million people aged 65 and over, in the USA alone, have Alzheimer’s dementia and the prevalence increases with age [1]. The global prevalence of dementia in 2021 is estimated at 55 million people with an additional 10 million cases occurring each year [2]. Due to the progressive nature of symptoms, dementia is a condition of major concern in health care and society locally, nationally, and cross-culturally at a global level.

On The Global Deterioration Scale [3], severe dementia is characterized by the inability to perform fundamental activities of daily living (ADLs), to communicate verbally, and by incontinence and severe psychomotor limitations [4]. Terminology for people in the later stages of dementia is varied, with “severe”, “advanced” and “end-stage” used interchangeably. The term “end stage dementia” (ESD) is used in this study. 

Further to functional losses, recognition of significant others can be impaired resulting in heavy dependence on caregivers [5]. End of life with dementia can be characterized by discomfort, pain, concurrent illnesses and burdensome interventions, resulting in reduced quality of life [5]. Behavioural problems tend to subside as passivity and apathy intensify and lethargy seems more common than agitation [6]. Degrees of impairment may vary considerably between individuals, e.g., loss of verbal communication but maintaining mobility; or lack of mobility and full dependence for ADL while maintaining verbal communication. For this reason, the focus of care should be to maximize overall quality of life by supporting individual needs [7]. 

### 1.2. The Role of Music Therapy for People with Dementia 

Music therapy is believed to be beneficial for people living with dementia at different stages of the disease. However, conclusive evidence is mixed due to many factors including heterogeneity among outcomes of interest, confusion surrounding the term “music therapy” and a lack of clarity regarding professional qualifications of interventionists [8]. Notwithstanding these caveats, the literature provides evidence for short-term effects of a wide range of music interventions in reducing behavioural and psychological symptoms of dementia (BPSD) [8]. In particular, there appears to be moderate quality evidence for the reduction of depressive symptoms [9]. While Domínguez-Chávez et al. [10] report positive effects of music interventions on cognitive functioning among people with dementia, this finding has not been substantiated by other reviews [11,12]. Reports on the effectiveness of music in reducing BPSDs tend to include individuals across the dementia spectrum, whereas those reporting effects on cognition generally do not include people with end-stage dementia. 

Literature from allied professions encourages professionals to identify areas of strength and residual abilities as well as limitations in the responsiveness of the person with dementia [13]. Understanding strengths and abilities helps to tailor levels of stimulation appropriate to individuals, thus developing appropriate management plans and interventions [13]. Music is an attractive medium for working with people in end-stage dementia because of its capacity for non-verbal communication and active engagement on many levels. Music interventions may be active or passive, and can be individualized based on preferences and response capabilities. 

Research on the use and benefits of music therapy among persons with end-stage dementia remains limited with a majority of studies examining cognitive function, behavioural and psychological functioning, and quality of life [14]. Despite the significant impairments associated with the disease, music interventions are accessible for people across the dementia spectrum and help to enhance cognitive functioning, promote social interactions and prosocial behaviours [15,16,17]. Research in the 1990s found that individuals in late-stage dementia could engage successfully with music through dancing and rhythm [18,19]. In a recent review of literature, Mercadal-Brotons [20] identified a number of responses to music among people in late stage dementia. These included increased alertness and engagement, active participation and the ability to sing familiar songs and to express positive feelings related to them when verbal skills still are present. Rhythm-based activities also elicited positive responses, especially when sensory stimulation was minimized (i.e., a cappella singing). Music interventions which are active, individualized and preference-based appear most successful in reducing BPSD. These findings are supported by Van der Steen et al. who note that people in advanced stages of dementia may be able to hum or play along with music [9]. 

### 1.3. Assessment for People with Advanced Dementia: Identifying Relevant Domains and Constructs

One of the difficulties in researching or evaluating music therapy with this population is the lack of measures that are relevant to the needs of the individual and that are sensitive to the population’s minimal responses. General assessments for people with dementia focus on a range of constructs across domains, including quality of life (QoL) [21]; anxiety [22]; pain [23], language and communication [24], and mild cognitive impairment in early dementia [25]. Standard cognitive assessments are used in the early stages of dementia as part of diagnostic evaluations, but their strong reliance on verbal skills may make them unsuitable for use among persons with advanced dementia due to floor effects. As many of these are screening tools, they also may lack sensitivity to incremental changes as a result of therapeutic intervention [26]. Proxy assessments are available for many versions of behaviour, QoL and depression scales, and are most suitable for those with late-stage dementia due to factors such as functional impairment or the effects of simultaneous use of multiple drugs [21,26]. There is considerable variation in the length and breadth of these tools; some require professional training to administer, some are better suited to research than to clinical practice, and scores on self and proxy versions may diverge, such as in the case of QoL measures [26]. Constructs such as quality of life and depression (or the emotional domain more broadly) are certainly pertinent to music therapy assessment with the person with end-stage dementia. However, single measures that focus on one domain can miss the holistic effects of music interventions with these minimally responsive individuals. Ray and Mittelman [27] note that music therapy can impact multiple outcomes, however, many standard tools may miss measuring vital components of the music experience, such as pleasure, experienced by the person with ESD.

### 1.4. Measures Available for Music Therapy with End-Stage Dementia

In the early 1990s, two music therapy assessments were developed specifically to observe and measure music responses among people with dementia. Clinical observations and relevant literature of the time provided the foundation for the development of the *Music-Based Evaluation of Cognitive Functioning (MBECF).* This assessment was designed to use music performance tasks to assess cognitive functioning among people with dementia [28,29]. The assessment has a structured protocol and has met acceptable psychometric criteria [28,30]. Around the same time, the *Residual Music Skills Test (RMST)* [31] also was developed to identify and measure music skills acquired through enculturation over one’s lifetime. These are referred to as “residual music skills,” and the test was designed primarily for music therapy clinicians to identify music skills that might be beneficial in communicating and interacting with individuals with Alzheimer’s dementia [32]. 

Both of these measures recognized the value of music tasks in the identification of cognitive and behavioural strengths among people with dementia. They were designed to provide information on functional capabilities that might not be assessed by non-music tools and to provide ways for music therapists to design evidence-based treatment interventions for this population. Studies cited by Baird and Sampson [33] indicate that separate neurological correlates may underly explicit and implicit memories for music, with implicit memory being less susceptible than explicit memory to the effects of dementia. Benhamou and Warren [34] indicate that preserved and acquired music playing competence may be preserved into severe stages of dementia. The MBCEF and RMST were designed to tap into these abilities, and also have the advantage of specific protocols and rating scales that include response options for people in advanced stages of dementia. 

More recently, Munk-Madsen [35] has provided a descriptive assessment of responsiveness in music therapy to help the therapist identify behaviours that may be measured quantitatively. Assessing responsiveness across six domains, the model is suitable for people with early to mid-stage dementia, but suggestions are offered for therapists who may wish to use the tool with individuals in later stages. The tool does not include a protocol or a scoring system but instead provides questions for therapists to consider as they conduct the assessment.

McDermott, Orrell & Ridder [36] have highlighted the need for developing music therapy-specific measures when working with people with dementia. Furthermore, identifying the characteristics of what ‘works’ and what is less effective in music therapy intervention may help to refine protocols for using music interventions with dementia populations [37]. However, the sensory and motor impairments that are prevalent in people with late to end-stage dementia challenge the use of standardized assessment protocols, reducing the validity of many test scores that result in a floor effect [13]. McDermott [34] also has noted that many of these tools are problem-focused, and that music therapy tools have the potential to better reflect a person-centered approach to dementia care. McDermott et al. [38] suggest that music therapy-specific measures may offer more sensitive tools for identifying subtle, clinically important changes within musical and social exchanges than other measures in people with dementia. For clinicians and researchers, a measure needs to identify the patient’s current status in order to evaluate change in a progressive or degenerative direction. In this way the measure helps to plan treatment that matches patient responsiveness. 

The Music in Dementia Assessment Scales (MiDAS) [39] was developed as a measure of how far people with moderate to severe dementia engage with music therapy. The MiDAS has five visual analogue items which evaluate an individual’s responses to music experiences on the constructs of Interest, Response, Initiation, Involvement and Enjoyment [36]. The measure is rated by both a carer and a therapist using Likert scales for each construct. A score of behavioural change is produced for each scale, providing a measure of responsiveness that can be tracked over time. The MiDAS has good psychometric properties and its reported benefits include ease of use and its utility as a treatment planning tool, offering insights into which patients may benefit on outcomes such as quality of life or reduced psychiatric symptoms [39]. However, it is limited in not measuring all the components of music therapy with people with dementia, e.g., across domains. It is reported as valid for use with people with moderate to severe dementia although having no protocol enables a therapist to modify the music experiences offered according to patient responsiveness and preferences. Thus it is potentially applicable across the full spectrum of dementia. 

### 1.5. Advantages of an Additional Tool for ESD

Music processing is complex, but recent research is shedding light on neurological mechanisms underlying these processes, and how they are affected by various types of dementia [34]. Continued development of validated, music-based instruments is necessary in order to more accurately identify and quantify both deficits and strengths and to use this information to develop and evaluate music therapy interventions. Greater understanding of the subtle changes in sensory and motor responsiveness in ESD would facilitate the development of individualized, evidence-based music therapy interventions. While psychometric data on both the MBECF and the RMST were promising, the lack of follow-up research on these tools leaves unanswered questions regarding their clinical usefulness among persons with ESD in a contemporary music therapy environment. Thus, the need for a more detailed assessment and evaluation of patient responsiveness to music therapy in end-stage dementia seems warranted.

### 1.6. Measuring Responsiveness in People Who Are Minimally Responsive

The Music Therapy Assessment Tool for Awareness in Disorders of Consciousness (MATADOC) has been established as a valid and reliable assessment of awareness in adults with prolonged Disorders of Consciousness (PDoC) stemming from acquired profound brain damage from trauma, illness or infection [40,41]. It was developed for use with patients who had emerged from coma, i.e., who show periods of wakefulness, but who remain unresponsive to their environment for more than 28 days [42]. Assessment of awareness in PDoC patients is complicated by the combination of severe physical, cognitive, sensory and communication impairments resulting from their brain damage. The MATADOC has particular value in developing individually tailored treatment for people who are minimally responsive based on data generated from the assessment [41]. The principal subscale rates responsiveness across sensory (visual and auditory), arousal and communication behaviours providing an assessment of awareness [40]. Subscales two and three rate responsiveness in functioning within the communication, motor, cognitive and psychosocial domains. These subscales identify behaviours that are emerging or have been retained, and thus contribute to goal-setting and intervention planning. The MATADOC can track small incremental changes typical in people who have complex needs due to a combination of cognitive, physical, sensory and communication. For people with PDoC, it is useful for both assessment and ongoing evaluation.

Although validated as a diagnostic measure for PDoC, the MATADOC provides a flexible protocol to include personalized and salient musical material. Furthermore, it provides a measure that does not have “floor effects”, which is a challenge when working with minimally responsive populations. The behavioural challenges of PDoC and ESD have many similarities across the cognitive, motor, communication, and psychosocial domains. Furthermore, the MATADOC assists with music therapy intervention planning by determining the components of music most likely to elicit responses. The central construct assessed by the MATADOC is awareness, which is not necessarily a construct of priority concern when working with people with ESD. However, the MATADOC protocol and documentation may be useful for clinical work with ESD due to its proven sensitivity with adults who have comparable complex needs stemming from acquired brain damage. An exploration of the MATADOC’s clinical utility with people with ESD demonstrating minimal responsiveness is warranted and will identify adaptations required to refine the MATADOC prior to testing its validity for people with ESD. To strengthen observations, concurrent data collection using a dementia-specific music therapy assessment may potentially reveal the relevance of the MATADOC items for this population. 

### 1.7. Objectives of This Study

This study aimed to explore the clinical utility of the MATADOC with a small sample of people with end-stage dementia, specifically to establish:(i)The clinical utility of the MATADOC protocol and of the 14 items of the MATADOC assessment for people with ESD using a multidimensional model for clinician judgements [43].(ii)Preliminary indications of reliability through test–retest (TRT) and inter-rater (IRR) administration with independent raters blind to each other’s ratings;(iii)Comparisons between clinical outcomes determined by MATADOC and the MiDAS.

Additionally, we hoped to identify MATADOC assessment items and protocol procedures that may benefit from modification for use with people with ESD. 

## 2. Materials and Methods

### 2.1. Design

A prospective study with repeated measures was used modelled on earlier studies testing the MATADOC with adults with PDoC [40,41] and examining its clinical utility with children with PDoC [44]. These studies compared ratings of live sessions between two MATADOC-trained raters for inter-rater reliability, and compared ratings of live sessions with video records of the same sessions from one rater for test–retest reliability. To examine its clinical utility and the modifications required for the new population of children, feedback on the protocol and documentation was collated from the raters who were experienced working with children with PDoC [44].

### 2.2. Setting 

Data collection took place over 8 months at a senior living community with assisted living providing activities and memory programs, caring for people living with early to end stage dementia in a principal city of a metropolitan area in the USA. Music therapy was already offered at the community center through contract with an independent certified provider. 

### 2.3. Participant Recruitment

There were two sets of participants: patient participants (senior living community residents with end-stage dementia) and professional participants (board certified music therapists). Through referrals from the Activities and Memory Program team, a convenience sample of people with ESD were recruited from the residents living in the senior living community over 7 months by the research team. Classification of “end-stage” dementia was determined through classification in residents’ medical records where available and by care facility staff. 

The inclusion criteria for patient participants included: adults with ESD as determined through residents’ clinical records or care team; who were identified as possibly benefiting from music therapy assessment or intervention; and could tolerate individual clinical contacts ranging from a minimum of 15 min to a maximum of 30 min in any single contact. Potential participants were excluded based on the following criteria: having a known moderate-profound hearing loss; had previously refused music therapy; were known to have adverse effects to music and/or music therapy; and were non-English speakers.

### 2.4. Procedure

Modelled on existing studies, two board certified music therapists served as data collectors (therapist-rater and observer-rater). Each data collector acted as interventionist to assigned patient participants and was responsible for delivering the MATADOC protocol [45] during four live clinical contacts. All clinical contacts were video recorded. This enabled data collection in both live and video conditions (therapist-rater), allowing for potential test–retest data. To enable inter-rater data, a second data collector (observer-rater) rated patient participants’ responses independently in either the live clinical contact or video observation. In this way, four inter-rater ratings were captured between the two data collectors (i.e., therapist-rater live rating compared to observer-rater live or video rating); and test–retest comparisons were enabled for four live-video comparisons (i.e., therapist-rater live compared to therapist-rater video rating). Ratings of video data for test–retest data were completed three to five weeks later to minimize risk of bias. Data collectors rated patient participants’ responses independently and remained blinded to each others’ ratings.

#### Intervention Protocol

Patient participants each received the MATADOC protocol weekly in four individual clinical contacts. The MATADOC protocol uses live (and occasionally recorded) music in a minimum of five procedures over 15–30 min duration (Table 1). Procedures emphasize salience (e.g., songs known to be meaningful to the patient and improvised songs using the patient’s name) and the visual presentation of musically related stimuli, e.g., instruments or pictures of the patient participant’s favorite musical artists [45]. The musical stimuli seek to elicit active behavioural responses (e.g., vocalization, localizing to sound source, responses to verbal commands) across cognitive, sensory, communication and motor domains in addition to optimizing the patient’s arousal. The MiDAS does not have a specific protocol and MiDAS data were collected pre and post the MATADOC protocol.

### 2.5. Measures

Concurrent data were collected with patient participants using the MATADOC assessment documentation post session and the MiDAS pre- and post-session for each of four clinical contacts. The MATADOC rates responsiveness in 14 items using Guttman (12 items) and binary (2 items) ratings across motor, communication, cognitive, emotional and sensory domains. The principal subscale produces a summed score ranging 0–10 providing an overall diagnostic outcome of awareness (Vegetative State or VS; Minimally Conscious State or MCS; or Emerged from MCS or EMCS). The MiDAS five items use visual analogue scales to collect data at two timepoints during each clinical contact: pre-session ratings are subtracted from during-session ratings providing one score of behavioural change [36]. MiDAS scores range from 0–500 per clinical contact [39]. In this study within-session ratings were completed at the end of each clinical contact, asking the data collector to think about the patient participant’s optimal responsiveness during the MATADOC intervention. Data collectors were trained in using the MATADOC to a recognized level of competency and had received preliminary training in using the MiDAS in line with recommendations by the scale’s developers [46]. Data collectors were blind to each other’s ratings.

Clinical utility of the MATADOC was evaluated post-data collection by therapist-raters using a 36-item questionnaire (categorical, ordinal and qualitative data) based on the multi-dimensional model for establishing clinical utility [43]. This questionnaire sought professional opinion of the appropriateness and acceptability of the MATADOC protocol procedures and relevance and usefulness of the MATADOC assessment items for people with ESD (see Appendix A).

### 2.6. Analysis 

Due to the small sample size typical of an exploratory pilot, analysis of the clinical utility questionnaire quantitative data were restricted to descriptive statistics. In such conditions, confidence intervals are a preferred method for describing the range of uncertainty in scores. Qualitative data were summarized to draw out clinician recommendations for adaptation. MATADOC and MiDAS data were examined for normality of distribution, prior to variance component analysis (VCA) of each measure’s dataset to examine how various components interacted and contributed to overall variance. 95% confidence intervals were obtained for the variance model. Intraclass correlation coefficients (ICCs) were used to examine the stability of the MATADOC over repeated measurements. Criterion validity was measured by correlating MATADOC and the MiDAS scores using Pearson’s correlation coefficient by rater, by session, by condition (live vs. video) and by score outcomes. Alpha levels for ICC and correlation were set to 0.05.

## 3. Results

Six patient participants were recruited who met the inclusion criteria, five of whom were female. Ages ranged from 75–98 years (mean 86.3, Table 2). Although the results of standardized diagnostic assessments for dementia were not available from participants’ clinical records, all participants were resident in a facility providing care in memory programs specifically for people with dementia. All participants were considered by the care team to be in the “end-stage” of the dementia trajectory given they were fully dependent for activities of daily living with minimal responsiveness to their environment. All patient participants received MATADOC sessions in addition to standard care offered at the facility. 

Data were collected for all six participants (MATADOC and pre- and post-session MiDAS) at four timepoints each by two raters (therapist-rater and observer-rater) to enable IRR. For TRT an additional rating at four timepoints was made by the therapist-rater who delivered the MATADOC protocol using video ratings of the live clinical contacts. This provided 72 data points in total and there were no missing data. Four participants scored MATADOC outcomes of MCS and two participants scored outcomes of EMCS (Table 2). Table 3 provides the summary statistics of MATADOC and MiDAS scores. One clinical utility questionnaire (see Appendix A) was completed by the therapist-rater for each patient at the completion of the MATADOC resulting in six data sets.

### 3.1. Clinical Utility

For the MATADOC protocol, the three-minute behavioural observation period pre and post the MATADOC session was rated as “too long” (66.6%). The Introduction to Musical Stimulus that starts the MATADOC protocol was rated as being appropriate in only 66.6% of cases with comments suggesting that the music needed adapting to be more stimulating or louder, and to introduce additional instruments, opportunities for interaction and a greater use of familiar music. The auditory stimuli procedure was rated as mostly relevant and appropriate (83%) and provided new information about the participants on 50% of occasions. The visual stimuli procedure was rated highly appropriate, and in most cases (83.3%) was relevant and provided new information. The verbal command and salient song procedures were rated as highly appropriate, relevant and provided new patient information in 100% of cases. The protocol was rated as improving working practice, providing new information on patient responsiveness and being useful in patient care. It mostly fitted therapists’ working practices (66.7%) but required adaptation of the environment in most cases. (Table 4).

For the items of the assessment documentation, all items were rated as “very relevant”. All items were rated as “very useful” with some exceptions: Item 1 “Responses to Visual Stimuli” and Item 8 “Vocalisation” were rated “somewhat useful” on one occasion (16.7%). Item 2 “Responses to Auditory Stimuli” was rated “somewhat useful” on two occasions (33.3%).

### 3.2. Reliability, Validity and Comparison of MATADOC and MiDAS Outcomes

The MATADOC has already been established to have good reliability and excellent validity for use with adults with PDoC when used by trained raters [36]. This study aimed to explore preliminary indications of its reliability and validity with people with ESD when used by trained raters. 

#### 3.2.1. Internal Consistency as a Measure of Reliability

To estimate reliability by measuring the internal consistency of the MATADOC, all 14 items of the scale were used as subcomponents. Internal consistency can be examined using omega, Cronbach’s alpha and Revelle’s beta, however omega is the best estimate [47]. Since the alpha calculation is sensitive to the number of items in the test it is potentially biased if used alone as an estimation of consistency. Therefore, in this study both the omega total and alpha were calculated for internal consistency, as suggested by Revelle and Condon [48]. Alpha and omega range from 0 to 1, with a higher value indicating a higher reliability in the MATADOC scale. A Cronbach’s alpha of 0 means that all sub-items are not correlated and are entirely independent from one another. If sub-items have high covariances, then alpha will approach 1. The higher the alpha, the more items have shared covariance and probably measure the same underlying concept. Coefficients at or above 0.80 are often considered sufficiently reliable to make decisions about individuals based on their observed scores, although a higher value, perhaps 0.90, is preferred if the decisions have significant consequences. The results found an alpha value of 0.802 and an omega value total of 0.865. Understanding that these results are based on a small sample, it suggests the MATADOC may have “good” internal consistency and be a sufficiently reliable measure when used with this new population. 

#### 3.2.2. Inter-Rater Reliability

Overall IRR was calculated by taking the ICC for a single rater (ICC2) randomly from the sample of “k” raters scoring each patient. The measure is one of absolute agreement in the scoring and removes mean differences between raters, but is sensitive to interactions of raters by patients. ICC2k reflects the mean of k raters. The MATADOC has an ICC of 0.35 with 95% CI (−0.004, 0.621; *p* < 0.005) for a single rater and 0.52 with 95% CI (−0.009, 0.766; *p* < 0.005) for k raters. Ratings in the video condition had an ICC of 0.54 with 95% CI (−0.002; 0.794; *p* < 0.02) for k raters than the live condition (0.404 with 95% CI (−0.282, 0.798; *p* < 0.05)). Due to the constraints from the size of data, TRT was calculated using the ICC for all sessions and 3 out of 4 sessions. Overall, for all sessions, the ICC was 0.54 (0.314, 0.759; *p* < 0.001) ranging from 0.44 (0.156, 0.702; *p* < 0.002) to 0.64 (0.387, 0.828; *p* < 0.001), with greatest agreement for consecutive sessions (either 1–3 or 2–4). (Table 5).

The mean MATADOC score for all 72 data points was 6.94 with SD 1.56 (an outcome of “MCS”). Scores ranged across MCS and EMCS, with no outcomes of VS being scored. VCA revealed that most of the total variance for the MATADOC data was attributed to the components of patient and rater, with “rater-session-patient” contributing further to variance. This may also reflect the small number of each in this exploratory study (Table 6).

#### 3.2.3. Criterion Validity

For validity, correlations between MATADOC and MiDAS scores were examined by rater, by session, by condition, by patient scores, with an overall correlation of R = 0.35 (*p* = 0.003) (Figure 1). Correlations for scores were significant to marginal significance for sessions 1–3 (ranging from R = 0.43 to 0.52 with *p* < 0.1) but not significant for the fourth session (R = 0.13, *p* = NS). Both live and video conditions showed moderate correlation (R = 0.33; *p* = 0.11 and R = 0.35; *p* = 0.016, respectively) between the two scores. Notably, correlations were good to very good for two participants only (#1: R = 0.75, *p* = 0.005; #3: R = 0.81, *p* = 0.0016) but otherwise were poor and not significant for the other the other four participants (Figure 1). 

## 4. Discussion

The clinical utility findings suggest that the MATADOC may be useful for assessing responsiveness to music therapy in people with ESD and particularly for informing treatment planning specific to an individual’s needs. However, both the protocol and assessment document require modifications to enhance its relevance for this new population. The pre- and post-session behaviour observation needs to be reduced from three minutes to two, as raters observed that participants became bored or anxious, particularly those patients who were more aware of their environment. The Introduction to Musical Stimulus at the start of the protocol may benefit from being more stimulating from the outset. Clinical utility data suggestions included playing with greater volume from the start of the music, building in opportunities for interaction, using a wider range of instruments and incorporating familiar melodies with personal salience to the client. All of these modifications could optimize music’s potential for enhancing arousal and better meet the needs of an older population whose hearing may be affected by age-related hearing loss. This contrasts with the PDoC protocol where the music builds element by element to layer the complexity of the auditory stimulus and identify those components that elicit responsiveness in patients whose awareness is unknown. Although the auditory item was rated as largely relevant and useful, rater responses indicated that the protocol should encourage the use of lower pitched instruments to compensate for high frequency age-related hearing loss with guidance for using greater volume throughout the protocol due to hearing loss. The most useful verbal commands were “point to the picture” (of a favourite artist) and “stop playing”. Choice-making tasks may be enhanced by asking clients to hold the objects, particularly in cases where language is believed to be compromised. This procedure may be enhanced overall by having a detailed procedure specifically around song choice using pictures. 

Given that people with ESD may be living in a care setting where the boundaries between mid, late and end stages of dementia are not well established, it is possible that the person being assessed may have greater mobility. Recommendations for revising the motor item included adapting the item to capture movement associated with agitation, e.g., getting out of a chair and moving around the room. Lastly, as agitation is a feature of mid- to end-stage dementia, the assessment would be improved by including an item to capture changes in this behaviour specifically. 

Observations about reliability and validity are less clear from the findings. Preliminary indications of reliability and criterion validity were always expected to be tentative given that data collection was planned with only a small number of participants and raters. The MATADOC’s reliability has already been established with PDoC patients [40,41], however we aimed to examine how it functioned with ESD patients when used by trained raters. Using alpha and omega estimates, the MATADOC’s internal consistency of all 14 items is good. This supports the clinical utility findings concerning the relevance and usefulness of the MATADOC items, and the measure’s utility for providing new information about a patient’s overall responsiveness. Notwithstanding the very small sample size of only six participants, the ICCs for TRT and IRR reliability were mostly above 0.5 which is considered ‘moderate’ reliability [49]. Moderate reliability with such a small sample, and one where all participants have very restricted function, certainly merits further exploration with a larger sample. Furthermore, the second major source of variance was the rater (see Table 5) which may be argued to be the case with any good measure. With only two raters and just six patients being rated, the influence of this variable is likely inflated and may well be reduced with more raters in a larger study. 

It is notable that no MATADOC scores fell within the VS range, suggesting that the sample was more responsive than people in VS. Music’s capacity to enhance arousal, cognition and emotions is often preserved even in advanced dementia [50]. Thus the results may reflect music’s known effects on arousal regulation in dementia [9] and abnormally enhanced cortical stimulation from music in some types of dementia [34]. Music is noted to be more potent than other stimuli in evoking personal memories in dementia (episodic musical memory), possibly as music-based memories are believed to be more emotionally salient [34] and less susceptible to the effects of dementia [33]. Alternatively, the behavioural repertoire of ESD may be too different than that of VS, being more comparable to behaviours associated with MCS. The degree to which items in the MATADOC tap into uniquely preserved musical abilities or abilities shared with other specific cognitive functions is a topic for future research. Construct validation studies should consider employing external reference standards for ESD that rate behaviours more closely aligned to MATADOC items (e.g., arousal; attention; initiation).

Score outcomes between MATADOC and MiDAS correlated in two cases only. Both measures are music-based (i.e., elicit responses through active music experiences) serving to guide treatment planning. However, the central constructs measured by each are different and the domains under which items are organized to measure each of these central constructs may explain the differences in the score correlations. The MATADOC assesses awareness through rating functioning across sensory, cognitive, behavioural and motor domains. The MiDAS assesses engagement with music through broader behaviours (response, initiation, involvement): MiDAS item correlations are highest for “Response” and “Involvement” (0.921, *p* < 0.001) [35] and these components might be understood to fall generally under the behavioural domain. MiDAS items with the poorest correlations are “Initiation” and “Enjoyment” (0.791, *p* < 0.001): these items might be better understood to fall under contrasting domains, being behavioural and emotional domains respectively. Thus, the MATADOC measures responsiveness through functioning, much like the MBECF [28] and the RMST [31], whereas the MiDAS is more concerned with identifying which patients are “likely to benefit in terms of improvement on other outcomes such as quality of life..and psychiatric symptoms” [39] (p. 1018). This difference implies a combined use of the MATADOC and MiDAS measures may offer a comprehensive assessment of function and quality of life to guide treatment. 

### Limitations

Missing information on cognitive status from standardized assessments for our included sample limits the conclusions we may draw from these findings. Future research needs to provide a full picture of cognitive function to optimize the potential uses of any new measure developed from the MATADOC for people with ESD. The use of descriptive statistics only due to the small number of patient participants further limits the assumptions that can be made from this exploratory study. To achieve narrower CIs for each component (rater-session-patient), a larger number of participants and raters is recommended. Increasing the number of raters may have provided more varied insights on the MATADOC’s clinical utility for this new population, particularly as the raters’ recommendations for adaptations to the protocol and items seemed to develop over repeated uses. With a larger sample size and more raters, statistical methods such as generalizability theory may identify more precisely the variance contributed to a total score by different components. Gaining clear diagnostic data of participants’ dementia status can be challenging given the progressive, degenerative nature of dementia and the nature of care settings. Future studies should ensure participants are screened to gain current disease status which will help to refine recommendations made for modifying the MATADOC for different stages of dementia. 

## 5. Conclusions

With modifications, the MATADOC protocol and assessment documentation may be a useful tool for assessing functioning and responsiveness to music interventions for people with ESD without the risk of floor effects. Testing the validity of the revised version with a larger sample will enhance its sensitivity further, providing an assessment for people with ESD to guide evidence-based intervention, help track changes over disease progression and support much needed research with this population. 

## Figures and Tables

**Figure 1 brainsci-12-01306-f001:**
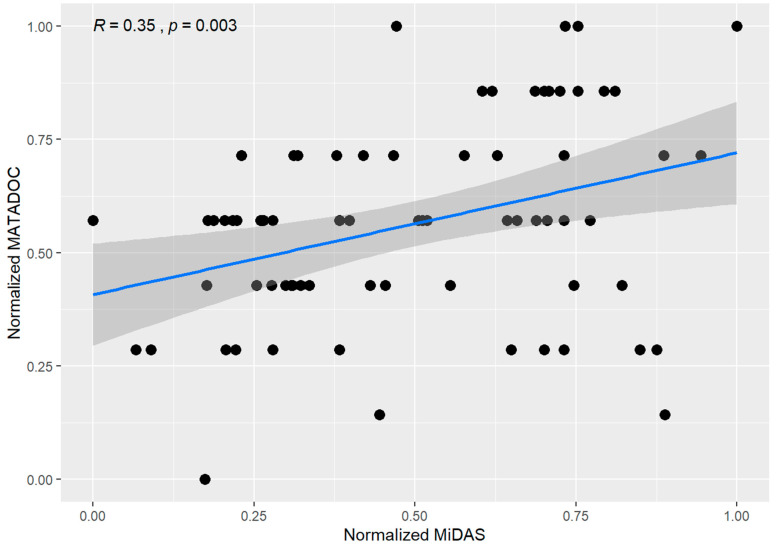
Correlations for MATADOC and MiDAS scores by patient, rater, session, and condition (live/video).

**Table 1 brainsci-12-01306-t001:** The MATADOC protocol.

MATADOC Protocol	
Procedure	Detail
3-min behavioural observation	No stimulation.
Procedure 1: Introduction of musical stimulus	Live music entrained to patient’s breathing. Music is introduced using single notes on guitar and gradually increased in intensity and complexity through the addition of musical component (timbre; rhythm; harmonic; melodic) in an improvised song that incorporates the patient’s name.
Procedure 2: Presentation of visual stimuli	Presentation of music-related visual stimuli (instruments; photos of favorite artists; album covers) in all visual quadrants to assess visual tracking.
Procedure 3: Presentation of auditory stimuli	Presentation of repeated single musical sounds (pitched or unpitched) on each sound to assess localization.
Procedure 4: Verbal command	Repeated presentation of a one-step verbal command.
Procedure 5: Presentation of salient familiar musical stimuli (live or recorded)	Presentation of one song known to be personally meaningful to the patient. Presented live unless an authentic rendition cannot be provided.
Procedure 6 (optional depdendent on patient responses): Procedures for items to inform goal-setting	Where the patient is showing responsiveness, procedures to build on any responsiveness exhibited by patient, e.g., vocalization; choice-making; instrument playing to assess purposeful physical movement. Omitted if patient is unresponsive.
3-min behavioural observation	No stimulation.

**Table 2 brainsci-12-01306-t002:** Participant sample with range of MATADOC outcomes.

Participant	Age	Documented Diagnosis	MATADOC Overall Outcome	Range of MATADOC Outcome Scores
1	86	NA	MCS	MCS-EMCS
2	98	NA	MCS	VS-MCS
3	91	Advanced dementia 2 years prior	EMCS	MCS-EMCS
4	75	Dementia/Alzheimer’s	MCS	MCS-EMCS
5	79	NA	MCS	MCS-EMCS
6	89	NA	EMCS	MCS-EMCS

**Table 3 brainsci-12-01306-t003:** Summary statistics of MATADOC and MiDAS scores.

	Mean Scores	Standard Deviation	Minimum	Maximum	Range	Standard Error
**MATADOC scores**	6.94	1.56	3	10	7	0.18
**Pre-MiDAS scores**	66.58	78.06	0	329	329	9.20
**Post-MiDAS scores**	281.94	130.32	36	496	460	15.36
**MiDAS change scores**	215.36	114.68	−16	449	465	13.52

**Table 4 brainsci-12-01306-t004:** Clinical utility results.

Procedure	Yes %	No %	NA/Absent %
**3-min observation period**			
Too long	66.7		
Just right	33.3		
**Introduction of musical stimulus procedure**			
Appropriateness	66.7	33.3	0
**Auditory stimuli procedure**			
Appropriateness	83.3	16.7	0
Relevance	83.3	0	16.7
Provided new information	50	33.3	16.7
**Visual stimuli procedure**			
Appropriateness	100	0	0
Relevance	83.3	16.7	0
Provided new information	83.3	16.7	0
**Verbal command procedure**			
Appropriateness	100	0	0
Relevance	100	0	0
Provided new information	100	0	0
**Salient song procedure**			
Appropriateness	100	0	0
Relevance	100	0	0
**MATADOC’s fit for therapist practice**			
Fits usual working practices	66.7	0	33.3
Challenges usual working practices	33.3	17.7	50
Improves usual working practices	100	0	0
Provided new information overall on patient responsiveness	100	0	0
Useful in patient’s care	100	0	0
Required special requirements of environment	83.3	0	16.7

**Table 5 brainsci-12-01306-t005:** Intra Class Correlations for Test–retest of MATADOC scores by session.

Session Number	ICC	Confidence Interval	*p* Value
All 4	0.54	(0.314, 0.759)	<0.001
(1,2,3)	0.64	(0.387, 0.828)	<0.001
(1,2,4)	0.46	(0.182, 0.722)	<0.001
(1,3,4)	0.44	(0.156, 0.702)	<0.002
(2,3,4)	0.61	(0.356, 0.813)	<0.001

**Table 6 brainsci-12-01306-t006:** Results of MATADOC variance component analysis examining interactions of component variables.

Variable	DF	SS	MS	VC	% Total	SD	CV [%]
**Total**	11.42	--------	---------	2.99	100	1.73	24.91
**Rater**	1	24.5	24.5	0.6	19.92	0.77	11.12
**Method**	1	0.84	0.84	0 *	0 *	0 *	0 *
**Session**	3	7	2.33	0.03	0.93	0.17	2.4
**Patient**	5	70.47	14.09	0.81	26.94	0.9	12.93
**Rater:Method**	1	1.33	1.33	0.01	0.17	0.07	1.04
**Rater:Session**	3	2.5	0.83	0 *	0 *	0 *	0 *
**Rater:Patient**	5	9.05	1.81	0.1	3.35	0.32	4.56
**Method:Session**	3	1.52	0.51	0.01	0.37	0.1	1.51
**Method:Patient**	4	5.08	1.27	0.27	8.9	0.52	7.43
**Session:Patient**	15	33.14	2.21	0.33	11.14	0.58	8.32
**Rater:Session:Patient**	15	15.26	1.02	0.64	21.41	0.8	11.53
**Error**	15	3.08	0.21	0.21	6.87	0.45	6.53

* VC set to 0.

## Data Availability

Data supporting the reported results is available upon request from the lead author.

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
