# Peer review of "Exploring the Clinical Utility of the Music Therapy Assessment Tool for Awareness in Disorders of Consciousness (MATADOC) with People with End-Stage Dementia"

_brainsci, 2022, doi:10.3390/brainsci12101306_

Round 1
Reviewer 1 Report
Thanks for the opportunity to review this manuscript aiming to preliminary investigate the clinical utility of MTADOC with end-stage dementia patients.
Overall, the theme of the manuscript is relevant and the methodology applied is acceptable.
The manuscript results are relevant and worth, in my opinion.
I have a few minor suggestions.
- the abstract lets the rest of the paper down; it should be more convincing and provide a few more details (i.e expand the idea of line 14 regarding the lack of ESD evidence based-intervention and the role of MT on this area; expand a bit more the needed modification in lines 24/25) and more clearly depict the objective and methods (the sentence starting in line 21 to line 24 is to confusing).
-Introduction is a bit too lengthy; page 1 line 38 add a space between 2021 and "is estimated"; there are other recent SR that could be included
- Material and Methods: the design could be further specified instead of just providing a reference (39); page 6 line 271: "three to five"?
- Discussion: page 11 line 425/426: justification for the specific suggestion of cutting down the observation period from 3 to 2 minutes (why 2 minutes and not 1 or 1 and a half?); the limitations section should be expanded.
Congratulation on the good work!
Reviewer 2 Report
This paper and study addresses a very important need for research with dementia. It presents a excellent overview of the challenge and what has already been developed in relation to music intervention research. This study itself is well conceived and documented. But there are a few questions and observations.
- Disorders of Consciousness - the definition of this very important construct for this study is assumed from previous research but needs some definition here indicating the normal broader use e.g. comas, locked in syndrome etc and why advanced dementia qualifies.
- Previous papers (2016) refers to PDoC. Clarify use of DOC vs PDOC here in relation to dementias? Is dementia not a prolonged form of DoC?
- in section 1.6 clarify the MATADOC has already been used with coma? TBI? Etc - ref Magee 2016?
- And in 1.6 suggest how this instrument works with traditional instruments such as MMSE - I.e., use in combination? At what score of MMSE do you add MATADOC, or something along this line. Or is it appropriate only when traditional instruments “hit the floor”?
- “MATADOC subscales two and three have utility primarily for goal setting and intervention planning.” - explain further?
- In conjunction with 2.4.2 it would be helpful to have an appendix or some form to provide actual, real world account and examples of how this assessment is conducted - something like a transcript. But could be an amalgam or fictional account but needs illustration to be clearly understood.
- 3.2. The assumption seems to be made that internal consistency is required or desired across the whole instrument. However, depending on factorial structure, this may not be the case. From the 2016 study The internal consistency of both Items 6 and 7 suggests that the within-item constructs are poorly related. Given the range of constructs included in each item, this seems accurate. I would suggest a discussion be added on factorial (components analysis) structure of the instrument and whether consistency on particular sections might be more appropriate.
- line 456 -457 - referencing needs editing
- Might the musical rhythmic presentation in items 6 and 7 have a neuromodulation effect and as such induce a response rather than elicit a response? The potential rhythmic Neuro modulator entrainment (neural driving) effects need to be acknowledged in the paper. May be particularly relevant in PDOC
- 3.2 in this section it would be helpful to be more precise as to type of reliability and especially type of validity is being discussed. Maybe use sub headers?
- Discussion: does using this music cognition (arousal from music?) test generalizes to cognition in general? Or is it limited to music processing and music- related or music stimulated cognition. Might there be some attention to what type of attention (awareness) is being observed with the test?
- Does this test primarily key into preserved mental ability unique to music and does music experience play a role in this? See the following references:
- Fornazzari, L.”Preservation of episodic musical memory in a pianist with Alzheimer’s Disease”. Neurology, 2006; 66: 610-614.
